# Design and Analysis of a Narrow Linewidth Laser Based on a Triple Euler Gradient Resonant Ring

Yikai Wang [1,*], Boxia Yan [2], Mi Zhou [2], Chenxi Sun [1], Yan Qi [2], Yanwei Wang [2], Yuanyuan Fan [2] and Qian Wang [2]

1 School of Integrated Circuits, University of Chinese Academy of Sciences, Beijing 100049, China; sunchenxi@ime.ac.cn

2 R & D Center of Optoelectronic Technology, Institute of Microelectronics, Chinese Academy of Sciences, Beijing 100094, China; yanboxia@ime.ac.cn (B.Y.); zhoumi@ime.ac.cn (M.Z.); qiyan@ime.ac.cn (Y.Q.); wangyanwei1@ime.ac.cn (Y.W.); fanyuanyuan@ime.ac.cn (Y.F.); wangqian4@ime.ac.cn (Q.W.)

* Correspondence: wangyikai@mail.ustc.edu.cn

**Abstract:** We designed a narrow-linewidth external-cavity hybrid laser leveraging a silicon-on-insulator triple Euler gradient resonant ring. The laser's outer cavity incorporates a compact, high-Q resonant ring with low loss. The straight waveguide part of the resonant ring adopts a width of 1.6 μm to ensure low loss transmission. The curved section is designed as an Euler gradient curved waveguide, which is beneficial for low loss and stable single-mode transmission. The design features an effective bending radius of only 26.35 μm, which significantly improves the compactness of the resonant ring and, in turn, reduces the overall footprint of the outer cavity chip. To bolster the laser power and cater to the varying shapes of semiconductor optical amplifier (SOA) spots, we designed a multi-tip edge coupler. Theoretical analysis indicates that this edge coupler can achieve an optical coupling efficiency of 85%. It also reveals that the edge coupler provides 3 dB vertical and horizontal alignment tolerances of 0.76 μm and 2.4 μm, respectively, for a spot with a beam waist radius of 1.98 μm × 0.99 μm. The outer cavity, designed with an Euler gradient micro-ring, can achieve a side-mode suppression ratio (SMSR) of 30 dB within a tuning range of 100 nm, with a round-trip loss of the entire cavity at 1.12 dB, and an expected theoretical laser linewidth of 300 Hz.

**Keywords:** narrow linewidth; external cavity; Euler gradient resonant ring; hybrid laser; SOI

## 1. Introduction

Narrow linewidth semiconductor lasers offer numerous advantages such as small volume and weight, high efficiency, long lifetime, narrow spectral linewidth, and excellent interactions. These lasers play a crucial role in various fields, including high coherence optical communication systems [1], dense wavelength division multiplexing (DWDM) systems [2], high-resolution optical sensing [3], precise clock timing [4], light detection and ranging (LiDAR) [5], and gravitational wave detection [6].

Wide laser tuning range, high laser output power, and narrow laser linewidth have traditionally been pursued by researchers in the field. One approach to accomplish these goals is through the utilization of the micro-ring resonator cavity (MRR). The MRR offers a wide free spectral region (FSR), which effectively extends the length of the resonator cavity. Additionally, the optical vernier effect can be employed to achieve a broad laser wavelength tuning range.

The performance of an external cavity laser is primarily evaluated based on two factors: tuning range and laser linewidth. The tuning range is directly determined by the optical vernier effect generated through the combination of resonant rings. On the other hand, the laser linewidth is influenced by the transmission loss of the waveguide, the overall optical length of the external cavity, and the laser's output power. It is worth noting that the transmission loss and the length of the external cavity are interrelated and mutually

influence each other. Increasing the length of the external cavity will inevitably lead to higher transmission losses.

After the pioneering work by Chu et al. [7], who successfully demonstrated the first tunable laser based on silicon photonics technology, numerous researchers have conducted extensive studies on resonant external cavities utilizing silicon waveguides and silicon nitride waveguides [8–11]. The theory behind the Q-value of the resonant ring and the wide-tunable external cavity laser has been extensively analyzed and studied [12]. Theory highlights that for external cavity lasers, the key factor in improving the Q-value of the resonant ring is the utilization of low-loss waveguides. Consequently, low-loss silicon nitride ($Si_3N_4$) waveguides have become the preferred material for external cavity lasers. The use of ultra-low loss silicon nitride waveguides can achieve extremely high Q-values for the resonant ring [13]. The combination of grating structure [14] or wedge structure [15] with a resonant ring can also effectively improve the Q-value of the resonant ring.

However, $Si_3N_4$ waveguides also have their drawbacks, including low thermo-optical sensitivity and high fabrication costs. On the other hand, silicon waveguides exhibit evident thermo-optical effects and offer certain advantages in terms of thermal tuning sensitivity. Nevertheless, silicon waveguides suffer from higher propagation losses under similar conditions. To address this challenge, Tran et al. [12] developed ultra-low loss silicon waveguides utilizing a shallow ridge waveguide structure, successfully reducing the transmission loss to 0.16 dB/cm. By incorporating a large three-ring external cavity, they achieved a wide tuning range of 110 nm and a laser linewidth of 220 Hz. However, as semiconductor lasers continue to progress towards increased integration, there is a growing demand for compact and space-efficient external cavity chips. In the case of circular resonant rings within the external cavity chip, if the shallow ridge waveguide technique mentioned by Tran et al. is employed, the minimum bending radius needs to be 600 µm. Such large footprints inevitably compromise the compactness of the external cavity chip. In order to solve this problem, based on Euler curve [16,17], this paper designs a compact, high Q-value Euler gradient resonant ring to replace the traditional circular resonant ring in the external cavity chip, which effectively improves the integration degree of the external cavity chip and shrinks the laser linewidth.

The output power of the laser is also an important factor that affects laser linewidth and overall performance. Increasing the optical power of the laser can effectively narrow the linewidth. In the context of coupling lasers with other components, such as laser plates, Tsuchizawa et al. [18]. proposed a low-loss mode field conversion structure (SSC). This structure effectively addresses the challenge of adapting the mode field between different components, enabling efficient edge coupling.

There are several forms of SSCs that have been explored, including layered polymer waveguide structures [19,20], multi-tip structures [21–23], nonlinear anticone structures [24], and subwavelength grating structures [25]. Among these options, a single nonlinear anti-cone structure may not be suitable for coupling with elliptic light spots from semiconductor optical amplifiers (SOA). The subwavelength grating structure is relatively simple to prepare but has a complex structure. On the other hand, the layered polymer waveguide structure offers excellent coupling performance but is challenging to fabricate.

The multi-tip structure is a simple and practical option that does not require complex fabrication techniques. It is particularly well-suited for the edge coupling of devices on the silicon-on-insulator (SOI) platform, and it has good compatibility with elliptical light spots.

In this paper, we analyze the mechanism of wavelength tuning in external cavity lasers and the factors that influence linewidth. We have designed an Euler gradient resonant ring to enhance the integration of the external cavity chip and to narrow the linewidth. To further improve the output power and reduce the laser linewidth, we propose a multi-tip edge coupler that minimizes coupling loss and enhances coupling tolerance. The structure of this paper is as follows: In Section 2, we theoretically analyze the wavelength tuning range of the external cavity laser and the multi-ring Vernier effect. In Section 3, we introduce the method for analyzing the intrinsic linewidth of the external cavity laser and discuss

the factors that influence the linewidth narrowing factor. In Section 4, we briefly introduce the basic structure of the laser, simulate and analyze the edge coupler and Euler gradient resonator ring, and calculate the expected linewidth of the laser. The thesis is summarized in Section 5.

## 2. Tuning Range and Side-Mode Suppression Ratio

A single resonant ring cannot achieve a wide tuning range, which can be addressed by using the optical Vernier effect [26,27]. The Vernier effect in micro-resonators is based on the principle that maximum power output is only possible when the maxima of the individual responses of resonators with unequal radii placed in series are aligned. This can produce a large free spectral range from two or more resonators with much larger radii. For a Vernier filter consisting of two resonators, the wavelength of the maximum must satisfy the following equation:

$$M \cdot FSR_1 = N \cdot FSR_2 = FSR_{\text{Tot}} \tag{1}$$

$FSR_{\text{Tot}}$ is the free spectral range of the combined resonators. $M$ and $N$ are integers, with $M = N + 1$ typically chosen in practical applications. Assuming symmetric coupling of the resonant rings, the FSR and complex amplitude transmittance of the resonant rings are given by:

$$FSR_m = \frac{\lambda^2}{n_g L_m} \tag{2}$$

$$t_m = \frac{-\kappa_m^2 \cdot e^{\frac{-j\varphi}{2}} \sqrt{a}}{1 - r_m^2 \cdot e^{-j\varphi_r} a} \tag{3}$$

$$t_{total} = \prod_m t_m \tag{4}$$

where $m$ is the number of the resonant ring; $\lambda$ is the central wavelength; $n_g$ is the group refractive index of the waveguide; $L_m$ is the perimeter of the resonant ring; $a$ is the loss factor of the light field transmitting in the resonant ring for one circle; $\varphi$ is the phase shift generated after the light field transmitting in the resonant cavity for one circle; $\kappa_m$ is the coupling coefficient, and the relationship between the coupling coefficient and the transmission coefficient $r_m$ is $\kappa_m^2 + r_m^2 = 1$; and $t_{total}$, in an ideal state, is the transmittance after synthesis.

However, when only a double ring is used to achieve a wide-tuning laser, a smaller free spectral range (FSR) difference between the two rings is required to provide a larger $M$ and $N$. This results in a significant increase in side-mode suppression ratio of the composite spectrum. The relationship between tuning range and side-mode suppression ratio is shown in Figure 1a. The primary reason for this phenomenon is that a larger $M$ parameter reduces the spectral difference between the two rings. Consequently, the side peak intensity near the main peak cannot be effectively suppressed in the composite spectrum due to insufficient spacing. Taking the first ring with a circumference of 400 μm as an example, when utilizing the double-ring Vernier effect, if a tuning range of 600 nm is desired, the side-mode suppression ratio of the spectrum would be reduced to 2.7 dB (Figure 1b), which clearly does not meet the design requirements for the external cavity.

In order to obtain a wider tuning range and a high side-mode suppression ratio, a third resonant ring can be added on the basis of the double ring to suppress the high side peak strength. The main peak of the third resonant ring is interleaved with the main peak of the double-ring spectral edge mode by setting the perimeter of the third resonant ring reasonably, so as to achieve the purpose of suppressing the edge mode intensity. When $FSR_3$ is set within the range of $FSR_1/2 < FSR_3 < FSR_1$, the peak valley of the third ring will overlap with the lateral peak, thereby suppressing the intensity of the lateral peak (Figure 1c). By introducing the third ring, the external cavity can simultaneously have a tuning range of 100 nm and a side-mode suppression ratio of 33 dB (Figure 1d).

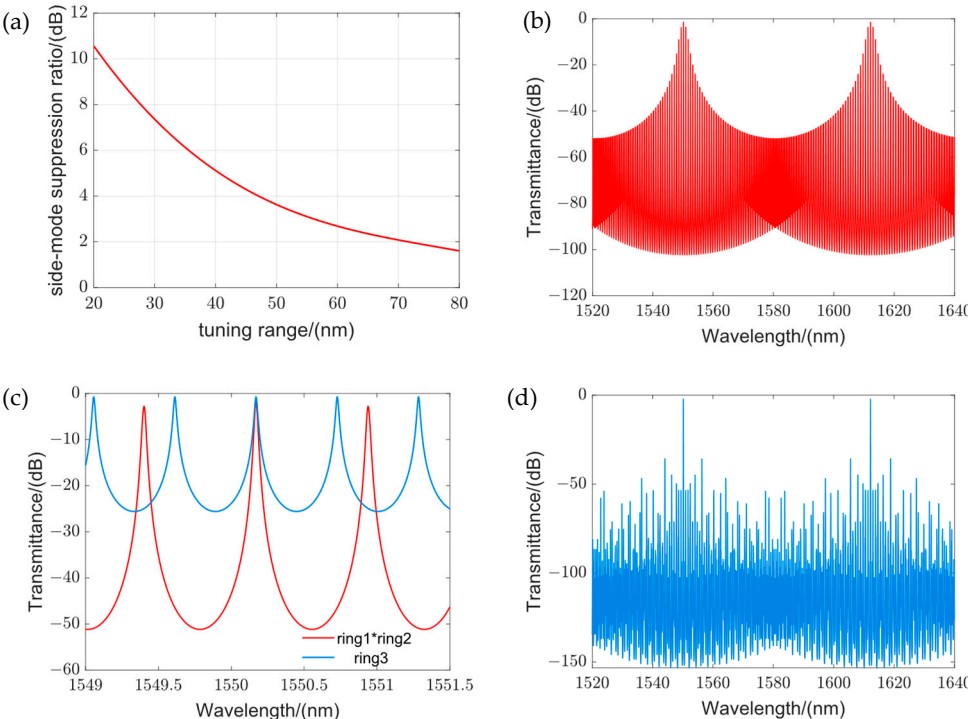

**Figure 1.** (**a**) Relationship curve between tuning range and side-mode suppression ratio; (**b**) double-ring vernier effect spectra with a tuning range of 60 nm, whose side-mode suppression ratio is only 2.7 dB due to its excessive M parameter; (**c**) a third ring is added on the basis of the double-ring vernier effect and to meet the requirements of $FSR_1/2$; (**d**) the intensity of the edge-mode resonance peak is effectively suppressed by the total transmission spectrum of the third ring after the addition of the third ring, and the side-mode suppression ratio is 33 dB.

### 3. Laser Linewidth Theory

The external cavity laser can be likened to a three-section laser cavity, as depicted in Figure 2a. The active segment (the yellow section) is located on the left side of the model, where the forward mirror has a reflectivity of $r_1$. The passive segment (the blue section) is situated on the right side of the model and includes the resonant ring structure, the passive routing waveguide, and the ring mirror. The left and right sides are interconnected via end-face couplers with a coupling loss $\beta$.

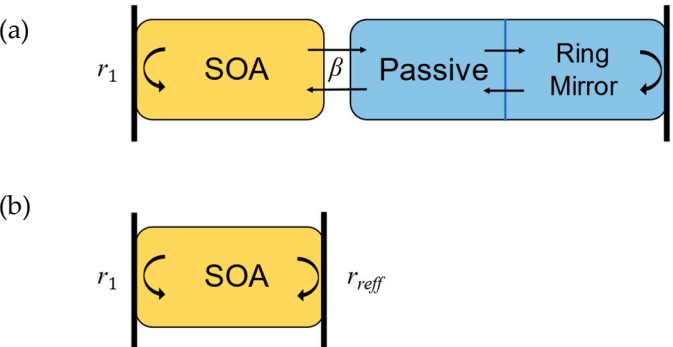

**Figure 2.** (**a**) The three-stage structure of an external cavity laser. (**b**) The simplified model of an external cavity laser.

For ease of analysis, the external cavity laser model can be simplified by equating the passive part shown in Figure 2a to the backward reflector of the SOA (Figure 2b). The

effective reflectivity $r_{eff}$ is the product of each transfer function of the passive part, given by the following equation:

$$r_{eff}(\omega) = \beta \cdot t^2_{passive}(\omega) \cdot r_{mirror}(\omega) \tag{5}$$

$$t_{passive}(\omega) = \exp(-\alpha_p L_p - j\beta_p L_p) \tag{6}$$

$$r_{mirror}(\omega) = r_{TSL} \cdot t^2_{total} \tag{7}$$

$$r_{TSL} = 2\kappa_{DC}t_{DC}\left[t^2_{DC}\left(e^{j\theta} + e^{j2\theta}\right) - \kappa^2_{DC}\left(e^{j\theta} + 1\right)\right] \tag{8}$$

In this context, $\beta$ denotes the coupling loss, which is approximately 1 dB. Specifics about the structure and value of the end coupler will be provided below. The transfer function of the wired waveguide is represented by $t_{passive}(\omega)$, wherein $\alpha_p$ and $\beta_p$ symbolize the electric field propagation loss in the silicon waveguide and the effective propagation constant, respectively. $L_p$ represents the total length of the routing waveguide. $r_{mirror}(\omega)$ is the equivalent mirror reflection function related to wavelength, with $r_{TSL}$ corresponding to the tunable Sagnac loop (TSL) mirror reflectivity. As depicted in Figure 3, TSL mirrors are installed behind the three micro-rings. Here, $\kappa_{DC}$ and $t_{DC}$ are the coupling parameters of the directional coupler, and $\theta$ represents the phase shift induced by the thermal phase modulator.

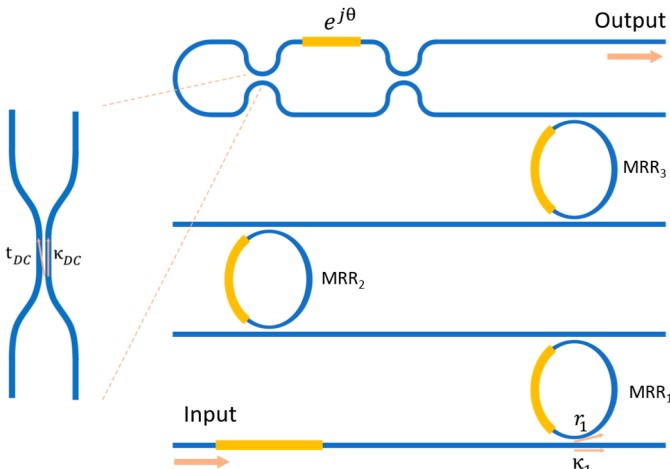

**Figure 3.** Structure of the three-ring external cavity. The laser sequentially passes through three resonant rings and is then reflected by the TSL mirror to form a resonance and proceed to output. Each resonant ring and the TSL mirror are equipped with a thermal phase shifter on the bus waveguide for phase alignment.

The mirror's reflectivity will fluctuate between 0 and 1 by adjusting the phase shift of the phase modulator. During laser operation, this reflectivity is optimized to harmonize with the laser's output power.

Upon equating the complex external resonator to the simplified resonator shown in Figure 2b, we can employ the formal method proposed by Patzak et al. [28], as well as Kazarinov and Henry [29]. Since the backward reflectance $r_{mirror}(\omega)$ is dependent on the laser frequency, the laser linewidth is substantially narrowed, attributable to the presence of the linewidth narrowing factor $F = 1 + A + B$. The formula for linewidth analysis used in this section is as follows:

$$\Delta\nu = \frac{\Delta\nu_0}{F^2} = \frac{\Delta\nu_0}{(1+A+B)^2} \tag{9}$$

$$A = \frac{1}{\tau_0}Re\left\{j\frac{d\ln r_{eff}(\omega)}{d\omega}\right\} = \frac{1}{\tau_0}\frac{d\varphi_{eff}}{d\omega} \tag{10}$$

$$B = -\frac{\alpha_H}{\tau_0} Im\left\{ j\frac{d\ln r_{eff}(\omega)}{d\omega} \right\} = \frac{\alpha_H}{\tau_0}\frac{d\ln\left|r_{eff}(\omega)\right|}{d\omega} \tag{11}$$

$$\Delta\nu_0 = \frac{1}{4\pi}\frac{v_g hvn_{sp}\alpha_{tot}\alpha_m}{P_0 K(\omega)}(1+\alpha_H^2) \tag{12}$$

$\tau_0 = 2n_g L_g / c$ is the round-trip time of photon in the active region, $\alpha_H$ is the linewidth enhancement factor, $v_g$ for effective group velocity of the active region, $h$ is Planck's constant, $v$ is laser frequency, $n_{sp}$ is the spontaneous emission constant, $\alpha_m = \frac{1}{L_g}\ln\left(\frac{1}{r_1|r_{eff}(\omega)|}\right)$ is mirror loss, $\alpha_{tot} = \alpha_i + \alpha_m$ is total loss, $P_0$ is the laser output power, and $K(\omega) > 1$ is the weight factor related to the transmittance of the output. Because increasing the reflectivity of the output will reduce the output power, $K(\omega) = 2$ is used in the following numerical calculations. The factor $A$ represents the ratio of photon transmission time between the gain endpoint and the external cavity waveguide. On the other hand, the factor $B$ denotes the strength of the optical negative feedback effect, reflecting the intensity of the phase change in the laser's external cavity. This coupling of phase and amplitude in the laser field serves to stabilize the laser frequency.

When the cavity is in resonance, the effective optical length of the ring resonator reaches its maximum, resulting in the factor $A$ also being at its peak. Correspondingly, the spectral curve resides at the resonance peak position where the rate of change is zero, $B = 0$. The presence of the factor $A$ and factor $B$ implies that a laser's outer cavity with a longer optical cavity length significantly narrows the laser linewidth. The higher the Q-value of the resonant ring in the multi-loop filter, the greater the length of the outer cavity that can be achieved, which in turn enhances the factor during resonance. Simultaneously, the reflection spectrum of a resonator with a high Q-value also exhibits a narrower full width at half maximum (FWHM). This condition results in a steeper spectral line slope at wavelengths deviating from the resonant peak, leading to an increase in the factor. From the above analysis, it is clear that the use of a higher Q-value micro-ring and an increase in laser power are pivotal in narrowing the linewidth. In this study, a graded resonant ring, optimized using an Euler curve, is employed to achieve a high Q-value while ensuring minimal external cavity loss. Moreover, a novel end coupler has been designed to enhance the inter-chip coupling efficiency, thereby boosting the laser power. Higher Q-values and higher laser power result in a narrower laser linewidth.

## 4. Laser Design

### 4.1. Laser Structure

The narrow linewidth external-cavity laser assembly, as shown in Figure 4, consists of a Si/InP gain chip (RSOA), an edge coupler, and a three-ring resonator mirror. The following description details this structure:

The Si/InP gain chip (RSOA) operates within a gain band of 1528–1568 nm (C-Band) and features a total internal reflection film that exhibits 95% reflectivity. On the opposite side, a 1000 m length is coated with a high-reflectance transmittance enhancement film to improve light transmittance. The gain chip employs a geometric technique, bending the ridge waveguide by 19.5°. This modification ensures that the incident light does not strike the chip's surface perpendicularly, further reducing reflectivity at one end of the chip. The exit end is coated with an antireflective film that effectively minimizes most back reflections.

Edge coupler: There is a significant mode mismatch between the spot size of the on-chip optical waveguide and that of the gain chip. Thus, a spot converter—an edge coupler in this case—is necessary for mode matching. In this study, we introduce an edge-face coupler using a multi-cone scheme, which not only provides higher coupling efficiency but also allows for greater alignment tolerance.

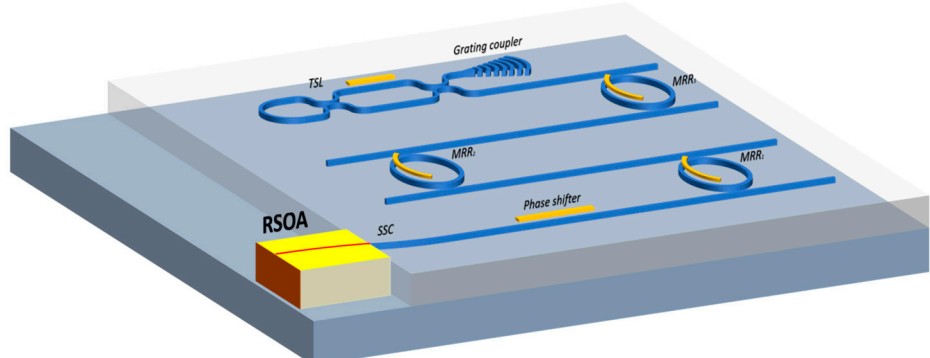

**Figure 4.** Schematic diagram of a narrow linewidth laser. ROSA and the outer cavity chip are connected by an edge coupler (SSC), and the output is connected by grating coupler after three resonant rings and TSL mirror. The bus waveguide, resonant ring, and mirror are all designed with thermal phase shifter for mode alignment.

Three-ring resonant mirror: The external cavity of the laser comprises three resonant rings and tunable Sagnac loop (TSL) mirrors. Each resonant ring and reflector section is equipped with a thermal phase modulator. A metal heating layer is vertically deposited on the oxide cladding of the silicon waveguide to form a micro-heater. This design leverages the thermo-optic effect for phase adjustment, facilitating intra-cavity mode alignment and reflectivity adjustment.

*4.2. Edge Couplers*

The laser is designed with a three-ring structure. The gain band ranges from 1528 nm to 1568 nm and the spot at the laser's exit surface has a beam waist radius of 1.98 μm × 0.99 μm.

It is apparent that a large beam area cannot be directly coupled with the waveguide mode of 220 nm by 450 nm. A spot size converter (SSC) is typically employed to enable efficient coupling between devices with dissimilar beam areas. However, a single-cone-structured SSC fails to achieve optimal mode matching for asymmetric elliptical beams. In this study, we designed a multi-tip edge coupling structure to enhance the efficiency of inter-chip coupling.

Figure 5a illustrates the device with a total length of 90 μm, consisting of a cone waveguide segment of $L_1 = 70$ μm and a transition waveguide segment of $L_2 = 20$ μm. Figure 5b depicts the cross-section diagram of the cone waveguide's right connection and the left side of the cone waveguide. $W_i$ and $W_j$ represent the width of the left and right waveguides, whereas $D_i$ and $D_j$ denote the distance of the left and right waveguides. $E_j$ signifies the side distance, and $H$ represents the waveguide height. The values of $D_i$ and $D_j$ depend on the minimum allowable spacing determined by the preparation process and the number of cone waveguides. Given the present state of the preparation process, the minimum spacing between waveguides is set at 200 nm, with $D_j$ fixed at 200 nm. To determine the coupling efficiency of various light spots, we calculate the TE optical mode of each waveguide cross-section by the MODE method. Subsequently, the coupling efficiency was computed using the following formula:

$$\eta_1 = \frac{\left| \int E_1 \times E_2 dA \right|^2}{\int |E_1|^2 dA \times \int |E_2|^2 dA} \tag{13}$$

$E_1$ and $E_2$ are the field intensity of the spot and the field intensity of the right side of the coupler.

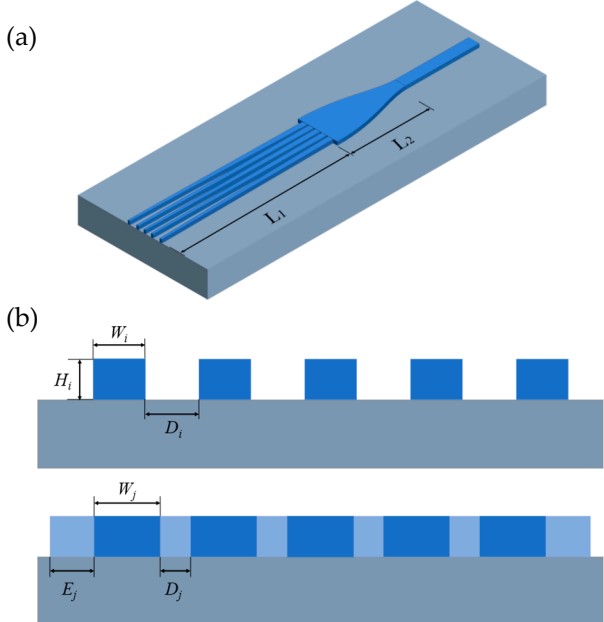

**Figure 5.** Multi-tip edge coupler: (**a**) drawing of the whole coupler; (**b**) section at the junction on the right side of the cone waveguide array and the transition waveguide (**top**) and left exit coupling cross-section (**bottom**).

We conducted parameter scanning to evaluate the mode coupling efficiency on both sides of the section, and the corresponding results are depicted in the Figure 6. In the case of the right section, we ultimately chose $W_j = 230$ nm and $E_j = 340$ nm, resulting in a maximum coupling efficiency of 99%. The values of $D_j$ and $W_i$ for the left waveguide need to correspond to the light spot that is being coupled. The figure presents the parameter scanning diagram of the coupling efficiency for the light spot with a beam waist radius of 1.98 μm × 0.99 μm. We selected $D_i = 540$ nm and $W_i = 234$ nm, resulting in a maximum coupling efficiency of 85%.

Additionally, this paper includes a simulation of the alignment tolerance for the designed edge coupler. The simulation shows that the multi-tip design provides a vertical 3 dB tolerance of 0.76 μm and a horizontal 3 dB tolerance of 2.4 μm for a spot with a beam waist radius of 1.98 μm by 0.99 μm. These tolerance values are significantly higher than those for an end coupler with a single-cone waveguide.

Table 1 presents a parameter comparison between the multi-tip edge coupler designed in this paper and other types of edge couplers. It is evident that, compared to other schemes, the edge coupler designed here can maintain high coupling efficiency with a shorter length, which aids in enhancing the integration of the laser. Furthermore, this design does not necessitate high precision in the fabrication process, which helps in controlling production costs.

**Table 1.** Performance comparison of end face couplers with different coupling schemes.

| Coupling Scheme | Coupling Loss (dB) | Polarization | Length (μm) | Comments |
|---|---|---|---|---|
| A 3D waveguide structure [20] | 2.8 | TE | 850 | Three-dimensional (3D) SU-8 taper, silicon inverted nanotapers. |
| | 4.1 | TM | | |
| Single inverted cone structure [30] | 0.2 | TE | 1000 | Low loss, broadband width, easy fabrication practical for mass production. |
| Subwavelength grating structure [25] | 0.63 | TE | 205 | Multiple grating combinations, integrated thermal phase modulators, phase compensation |

**Table 1.** *Cont.*

| Coupling Scheme | Coupling Loss (dB) | Polarization | Length (µm) | Comments |
|---|---|---|---|---|
| Layered polymer waveguide structures [19] | 0.36 | TE | 300 | Thermal oxidation process, good TM mode support |
| | 0.66 | TM | | |
| This work | 0.7 | TE | 90 | Compact, multi pointed, high coupling tolerance, low process requirements |

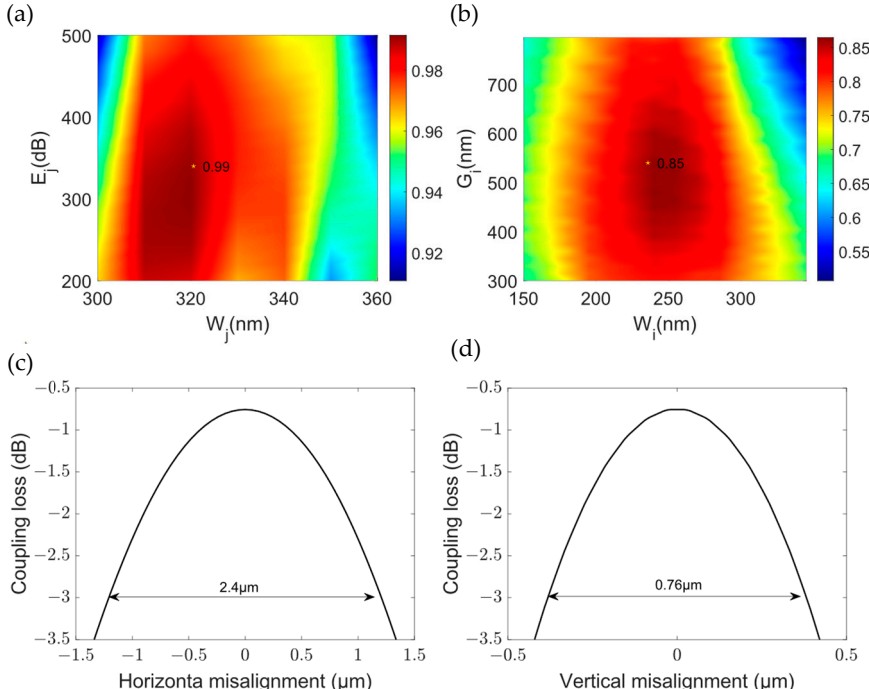

**Figure 6.** (**a**) Parametric scan of the coupling efficiency at the left section of the waveguide. (**b**) Parameter scan of coupling efficiency at the right side of the waveguide. (**c**) Horizontal alignment tolerance of the end coupler. (**d**) Vertical alignment tolerance of the end coupler.

### 4.3. Euler Gradient Ring Design

According to the analysis of laser linewidth in Section 3, a high Q-value of the resonant ring can enhance the spectral characteristics of the external cavity mirror. The Q-value of a symmetrically coupled resonant ring can be calculated using the full width at half maximum (FWHM) of the transmission spectrum.

$$Q = \frac{\lambda_0}{FWHM(\lambda)} \tag{14}$$

$$FWHM(\lambda) = \frac{\lambda_0^2}{\pi L n_g} \frac{1 - ar^2}{\sqrt{a(1 - \kappa^2)}} \tag{15}$$

Equation (15) reflects three factors that affect the resonant ring Q-value: resonant ring perimeter $L$, coupling coefficient $\kappa$, and loss factor $a$.

To minimize transmission loss, one approach is to increase the width of the waveguide. By doing so, the loss during transmission can be reduced. Regarding bending loss, the use of nonlinear curves can be employed to optimize this aspect. Nonlinear curves can help mitigate the bending loss, thereby enhancing the overall efficiency of the resonant ring.

Payne–Lacey theory [31] suggests that the unit transmission scattering loss of straight light waves caused by sidewall roughness can be expressed as:

$$\alpha = 4.34 \frac{\sigma^2}{k_0 d^4 n_2 \sqrt{2}} g(V) \cdot f_e(x, \gamma) \tag{16}$$

Among them, the unit for scattering loss $\alpha$ is given in (dB/cm). Here, $k_0 = \frac{2\pi}{\lambda}$ is the wave vector in vacuum. In general, the roughness of the waveguide sidewall can follow two types of distributions: exponential or Gaussian [32]. Formula (16) can be described as follows:

$$\alpha \leq \frac{\delta^2}{k_0 d^4 n_1} m \tag{17}$$

The coefficient m is 0.48 for an exponential distribution and 0.76 for a Gaussian distribution. The relationship between the transmission loss, caused by waveguide roughness, and half of the waveguide width ($d$) is depicted in Figure 7.

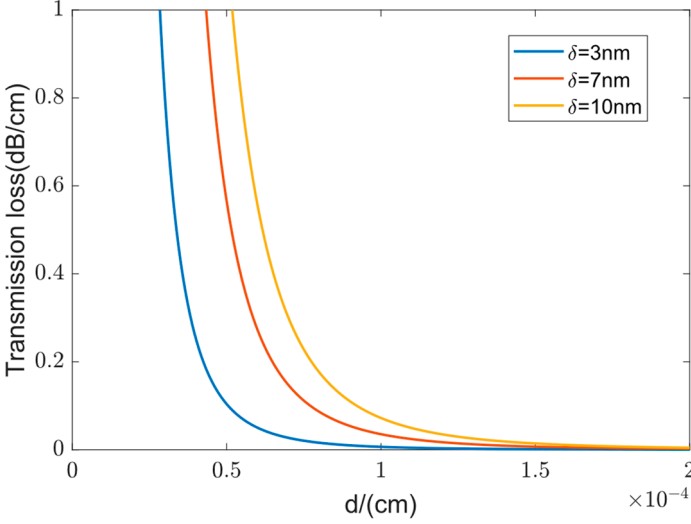

**Figure 7.** The relationship between transmission loss and half waveguide width.

Based on the analysis from the figure above, the scattering loss of the optical waveguide is inversely proportional to the width of the waveguide's cross-section. Considering the ordinate, when the width of the waveguide is constant, the unit transmission loss of the optical waveguide is directly proportional to the square of the sidewall roughness $\delta$. For SOI rib waveguides, scattering loss constitutes the main component of the unit transmission loss. Additionally, as depicted in Figure 7, the scattering loss remains essentially unchanged beyond a half waveguide width of 1.2 µm. Taking into account that excessively wide waveguides are prone to multimode operation, a waveguide with a width of 1.6 µm for transmission is a favorable choice.

Considering both transmission loss and bending loss, this paper proposes the design of an Euler gradient resonant ring to replace the original uniform circular resonant ring. The Euler gradient resonant ring utilizes an optimized width and nonlinear curves to minimize losses and enhance the performance of the resonant ring system.

The resonant ring is designed using an Euler curve, as depicted in Figure 8. This design results in an asymptotic bending waveguide profile, determined by the Euler curve [16,17] and governed by the equation $\frac{d\theta}{dL} = \frac{1}{R} = \frac{L}{A^2} + \frac{1}{R_{max}}$. In this equation, $R$ represents the radius of curvature of the curve, $L$ is the arc length of the curve from the initial end of the asymptote to the set point, and $\theta$ signifies the curve's angle. $A$ is a constant value that determines the total length of the curve and the minimum bending radius. As the arc length increases, the curvature radius of the curve gradually decreases from $R_{max}$ to $R_{min}$ at a 90° angle. Additionally, the width of the curved waveguide transitions from

1600 nm in the transmission region to 450 nm in the coupling region. The 450 nm end of the curved waveguide connects to a rectangular waveguide with a length of $L_x$, serving as the coupling area. Two 90° Euler gradient waveguides, along with the coupled waveguides, form a 180° Euler gradient U-shaped bending structure, and its effective bending radius is $R_{eff}$. Simultaneously, a linear transmission waveguide, 1600 nm wide and $L_c$ units long, is positioned between the two 180° bends. This, together with the two U-shaped curved waveguides, constitutes the complete structure of the resonant ring. The design of the curved waveguide combines the curved waveguide and taper waveguide for low-loss bending transmission and the ability to alter the mode spot size. This achieves the low-loss and stable transmission of a single mode, greatly enhancing the integration degree.

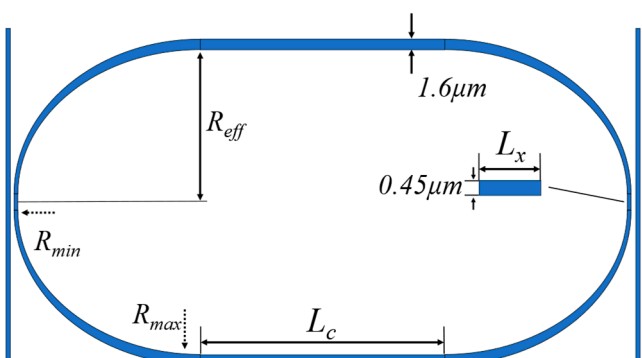

**Figure 8.** Schematic diagram of an Euler gradient resonant ring structure.

Using a 450 nm single-mode waveguide in the coupling region ensures efficient coupling and single-mode transmission. The single-mode nature of the waveguide helps maintain the integrity of the transmitted signal. Additionally, the Euler gradient design facilitates adiabatic transmission during waveguide width transitions, ensuring smooth and efficient signal transfer without significant loss or distortion.

To reduce the overall propagation loss of the resonant ring, a wide waveguide with a width of 1600 nm is utilized. According to simulation results and relevant literature [32,33], increasing the waveguide width from 450 nm to 1600 nm can reduce the transmission loss from 1.5 dB/cm to 0.3 dB/cm or even lower, thereby improving the overall performance of the resonant ring.

The large bending radius at the connection helps to minimize mode mismatch loss at the connection between the straight waveguide and the curved waveguide. The larger bending radius reduces the mismatch between the mode field distributions of the two waveguides, resulting in less loss during the mode transition.

Inserting a 1.6 μm wide transmission waveguide between two 180° curved waveguides allows for the alteration of the resonant ring's perimeter. Changing the perimeter provides control over the free spectral range (FSR) of the resonant ring, which enables the tuning of the resonant frequencies and facilitates applications such as filtering or wavelength selection.

Figure 9 is presented to illustrate the relationship between the resonant ring's perimeter and coupling mismatch loss. To achieve a mode excitation ratio of $TE_0$ to $TE_1$ below $-40$ dB, it is necessary for the radius $R_{max}$ to exceed 700 μm. Consequently, the value of $R_{max}$ is ultimately set to 900 μm in this study.

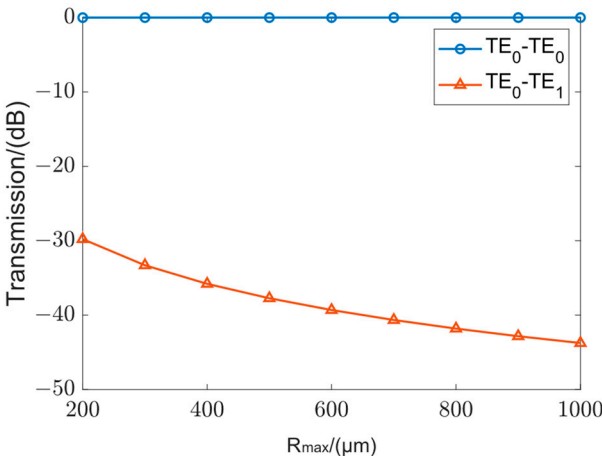

**Figure 9.** The relationship between the transmittance of $TE_0$–$TE_0$ and the excitation ratio of $TE_0$–$TE_1$ as a function of $R_{max}$.

Considering both the size and radiation loss of the resonant ring, the $R_{min}$ of the resonant ring should be as small as possible. Experimental results indicate that radiation loss can be disregarded for values of $R_{min}$ greater than 10 µm [34]. In this specific research, the values chosen are $A = 32$ and $R_{min} = 18$ µm, allowing for the smallest bending radius while still neglecting radiation loss. Given these parameters, the total length of the 90° Euler gradient bending is determined to be $L = 55.6$ µm. Meanwhile, the effective bending radius ($R_{eff}$) of this structure is only 26.35 µm. Compared to other articles [8–12], the radius of several hundred micrometers is a significant improvement.

After the parameters of 90° Euler gradient bending are determined, the U-shaped curved waveguide consisting of the waveguide is simulated by FDTD (as shown in Figure 10) at the wavelength of 1550–1660 nm. The bending loss of the waveguide is 0.014 dB, which effectively reduces the bending loss of the resonant ring. At the same time, the excitation ratio of the $TE_0$–$TE_1$ mode of the Euler gradient bending reaches −40 dB.

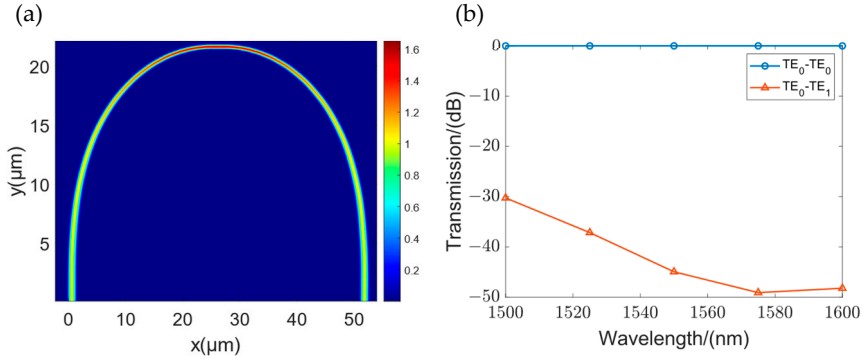

**Figure 10.** (**a**) FDTD simulation of curved waveguides. (**b**) The relationship between the transmittance of $TE_0$–$TE_0$ and the excitation ratio of $TE_0$–$TE_1$ as a function.

The Q-value of a resonant ring can indeed be improved by increasing its perimeter. However, it is important to consider that a larger perimeter can also lead to higher waveguide transmission losses. Taking into account the waveguide transmission losses, a perimeter value of 600 µm is selected for the first ring. This value can achieve a high resonant ring Q-value while ensuring low external cavity loss.

Using Formula (14), the Q-value of a 600 µm circumference Euler gradient microring is calculated to be $4.3 \times 10^4$. However, if a high-loss traditional narrow waveguide resonant ring is used, the Q-value would decrease to $3.9 \times 10^4$. Meanwhile, a ring with a circumference of 600 µm has a radius of 95 µm. In contrast, the effective bending radius of

the Euler gradient resonant micro-ring designed in this article is significantly smaller at only 26.35 μm.

Accounting for the waveguide width's impact on the group refractive index of the waveguide group, adjustments must be made from Formula (15) to (16) to determine the overall FSR of the Euler resonant ring:

$$FSR_m = \frac{\lambda^2}{2\overline{n}_g L_u + 2n_g L_c(m)} \tag{18}$$

$$L_m = 2(L_u + L_c(m)) \tag{19}$$

$\overline{n}_g$ is the average group refractive index of the U-shaped curved waveguide, $L_u$ is the length of the U-shaped waveguide, $n_g$ is the group refractive index of the straight waveguide, $L_c(m)$ is the length of the straight waveguide segment, and $L_m$ is the total length of the resonant ring, where $m$ is the number of the resonant ring. According to Section 2, when $L_1 = 600$ μm, in order to achieve the tuning range of 100 nm, the total length of the second ring is $L = 606$ μm and the length of the third ring is $L_3 = 778$ μm.

The power coupling coefficient dictates the extent to which light circulates within the resonant ring. According to the Formula (15), a smaller coupling coefficient can elevate the Q-value of the resonant ring. However, the presence of a loss factor means that an excessively small coupling coefficient can result in the additional loss of resonant light, consequently diminishing the intensity of the resonance peak. For external cavity lasers, excessive external cavity loss will significantly affect the laser output power. To avoid introducing large external cavity reflection losses, we choose $\kappa_1^2 = \kappa_2^2 = \kappa_3^2 = 0.1$.

### 4.4. Theoretical Linewidth Calculation for Lasers

The bus waveguide, used to connect the resonant ring across all levels, employs a 450 nm wide ridge waveguide with a length ($L_p$) of 2000 μm. Its loss calculation is based on the typical value of a 450 nm wide ridge waveguide, which is 1.5 dB/cm. Simultaneously, the round-trip loss of the resonant ring, the coupling loss between the RSOA and the external cavity, and the output power of the primary gain chip are incorporated into the calculations, with the output power ($P_0$) set to 10 mW. The perimeters of the three rings are $L_1 = 600$ μm, $L_2 = 606$ μm, and $L_3 = 778$ μm, with coupling coefficients ($\kappa_1^2 = \kappa_2^2 = \kappa_3^2$) set to 0.1. The external cavity reflectance spectrum is plotted using Formula (4), as illustrated in Figure 11a.

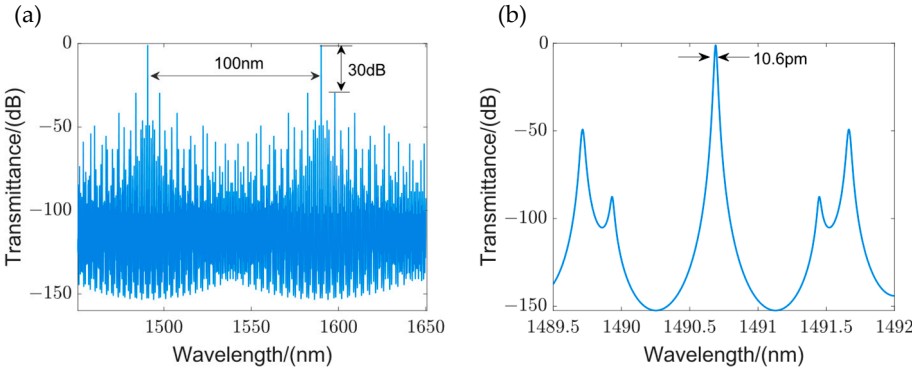

**Figure 11.** (**a**) Reflection spectra of the three-ring external cavity. The spectrum has a tuning range of 100 nm and a side-mode suppression ratio of 30 dB. (**b**) Enlarged image at resonance peak; FWHM is 10.6 pm.

The synthesized spectrum of the external cavity with the gradient Eulerian resonance ring indicates a tuning range of 100 nm and a side-mode suppression ratio of 30 dB. As shown in Figure 11b, owing to the gradient Eulerian resonance ring, the spectral resonance peak of the external cavity displays a normalized intensity of −1.12 dB, and its full width

at half maximum (FWHM) is only 10.6 pm. If a standard resonant ring with the same circumference is used, normalized intensity will decrease to −2.4 dB.

The method proposed in Section 4 is employed to analyze the linewidth of the outer cavity with a three-ring configuration. To highlight the benefits of the low transmission loss associated with the wide waveguide utilized in this study, both the linewidth narrowing factor and the linewidth were calculated for transmission losses of 1.5 dB and 0.3 dB, respectively.

The *B* factor attains its maximum value at the rising resonance frequency, while the *A* factor curve exhibits symmetry and reaches its maximum value at the resonance frequency. Consequently, the *F* factor also reaches its peak value at a slightly deviated resonance frequency. Figure 12a illustrates the results.

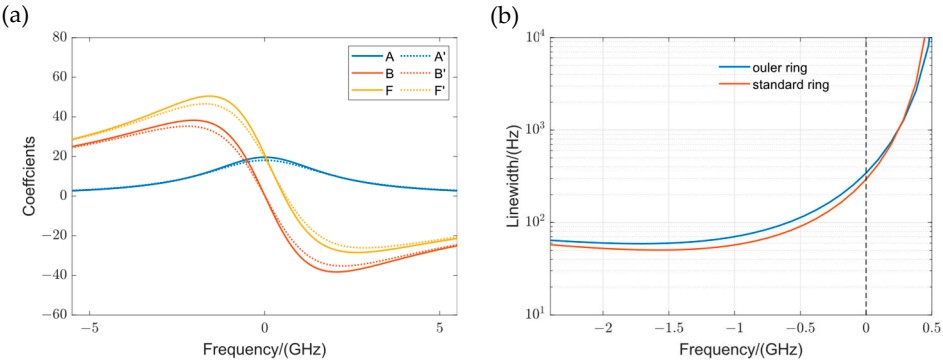

**Figure 12.** (**a**) Calculated values for coefficients A, B, and F. The solid and dashed lines represent the external cavities using Euler gradient resonant rings and standard resonant rings, respectively; (**b**) estimated Lorentzian linewidth as functions of frequency detuned from the dual ring.

It can be observed that, compared to the use of conventional micro-rings, the linewidth narrowing factor for the resonant outer cavity employing Euler gradient micro-rings has been improved to varying extents. The maximum F-factor for a three-ring outer cavity using a standard narrow waveguide resonant ring is 45, whereas the maximum F-factor for a three-ring outer cavity using an Euler gradient resonant ring is 50.

Upon inputting data into Formula (9), the laser's linewidth curve is derived and presented in Figure 12b. The three-ring resonator, which utilizes an Euler gradient resonant ring, is anticipated to have a linewidth narrower than 300 Hz at resonance. In contrast, the linewidth of a standard circular ring is 400 Hz, and this calculation is based on the assumption of equal output power. In reality, the higher loss of the standard circular ring results in reduced laser power, which would further decrease the expected linewidth.

## 5. Conclusions

As a benefit of the Euler gradient resonant ring proposed in this paper, the utilization of the three-ring vernier effect enables us to achieve a wide tuning range while maintaining a low side-mode suppression ratio. The introduction of the new Euler gradient resonant ring not only reduces the transmission loss of the resonant ring at the same technological level but also enhances the overall compactness of the optical external cavity chip with a bending radius of 26.35 µm. The designed Euler triple ring external cavity laser can achieve a theoretically narrow linewidth output of 300 Hz, while also possessing a tuning range of 100 nm and a side-mode suppression ratio of 30 dB.

As photonic integration continues to advance, the integration and miniaturization of each device within the photonics circuit become imperative. It is essential to explore the miniaturization of optical devices such as resonant rings and modulators. It is anticipated that in the future, with the advent of higher-precision waveguide preparation processes or novel materials with superior performance, the proposed Euler gradient micro-ring highlighted in this paper can further enhance the performance of external cavity lasers and offer fresh insights for optimizing the direction of the resonator design.

**Author Contributions:** Conceptualization, Y.W. (Yikai Wang) and B.Y.; methodology, Y.W. (Yikai Wang) and M.Z.; validation, Y.W. (Yikai Wang), C.S. and B.Y.; resources, Y.W. (Yanwei Wang) and Q.W.; data curation, Y.W. (Yikai Wang); writing—original draft preparation, Y.W. (Yikai Wang); writing—review and editing, Y.W. (Yikai Wang), B.Y. and Y.F.; project administration, Y.Q. All authors have read and agreed to the published version of the manuscript.

**Funding:** This research was funded by National Key R&D Program of China, grant number 2023YFF0718702.

**Conflicts of Interest:** The authors declare no conflict of interest.

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
