# Peer review of "Design and Analysis of a Narrow Linewidth Laser Based on a Triple Euler Gradient Resonant Ring"

_photonics, doi:10.3390/photonics11050412_

Round 1
Reviewer 1 Report
Comments and Suggestions for Authors
The Authors have designed a narrow linewidth external cavity hybrid laser based on an SOI triple Euler gradient resonant ring. The reported results were achieved through FDTD simulations. The proposed device shows great promise, although significant revisions are needed before publication.
Here are my comments:
1. In the Introduction, the authors discuss the relevance of a narrow linewidth. However, the literature on this topic should be expanded (see, for example, "Chemically etched ultrahigh-Q wedge-resonator on a silicon chip," Nat Photonics 6(6), (2012). "Ultralow 0.034 dB/m loss wafer-scale integrated photonics realizing 720 million Q and 380 μW threshold Brillouin lasing," Opt Lett 47(7), (2022). "Comprehensive mathematical modelling of ultra-high Q grating-assisted ring resonators," Journal of Optics 22(3), 035802 (2020)).
2. The Vernier effect is used to enlarge the FSR. However, this choice should be justified in alignment with the target application. The authors need to emphasize the desired performance criteria, such as FSR, linewidth, compactness, etc.
3. Figures 2 and 3 need improvement, mainly in terms of caption.
4. A 2D analysis has been conducted. However, this might affect the rigor of the results. Please provide commentary on this aspect.
5. The authors have employed a multi-tip-edge coupler. However, several solutions with potentially better performance have been proposed in the literature (see, e.g., "Silicon Nitride Spot Size Converter With Very Low-Loss Over the C-Band," IEEE Photonics Technology Letters 35(22), (2023), “CMOS-compatible spot-size converter for optical fiber to sub-µm silicon waveguide coupling with low-loss lowwavelength dependence and high tolerance to misalignment,” Proc. SPIE, 9752, 2016, “Ultra-lowloss inverted taper coupler for silicon-on-insulator ridge waveguide,” Opt. Commun., 283(19), (2010), “Silicon knife-edge taper waveguide for ultralow-loss spot-size converter fabricated by photolithography,” Appl. Phys. Lett., 102 (10), (2013)). Please discuss this and compare the achieved results with the existing literature.
6. Section 4.C requires extensive editing. Specifically, the authors need to justify the selection of the Euler grating ring over the standard one. This comparison should include both ER and Q-factor. Furthermore, a detailed description of parameters such as Rmax, Rmin, assumed propagation losses, and waveguide widths (450 nm and 1600 nm) should be provided. Additionally, the authors mention that Rmin > 10 um. Please provide evidence for this claim.
Reviewer 2 Report
Comments and Suggestions for Authors
The author designed a narrow linewidth external cavity hybrid laser based on a silicon-on insulator triple Euler gradient resonant ring. Good simulation result is confirmed, it would be better to add some experimental validation, minor revision of language is needed.
The simulation part is relatively comprehensive, but the manuscript is poorly written. The authors have many grammatical errors, mainly focusing on using articles and prepositions, which appear throughout the paper. There are also some misspellings of words, such as the horizontal coordinates in Figure 6. Paper writing norms need to be improved. Spaces are needed between numbers and units in the paper. The case needs to be handled, and Chinese punctuation cannot appear in English. There are also errors in paper citations, such as citation 12. There are even some repetitive words on page 2. The position of the image needs to be adjusted. Some pictures cover the header, such as Figures 3, 5, and 8. Meanwhile, Figure 10 obscures the contents of Figure 9. Comments on the Quality of English Language
minor revision is obliged
Round 2
Reviewer 1 Report
Comments and Suggestions for Authors
The Authors have modified the manuscript according to the Reviewer suggestions.